# Comparison of *Ips cembrae* (Coleoptera: Curculionidae) Capture Methods: Small Trap Trees Caught the Most Beetles

**Karolina Resnerová [1,*], Jaroslav Holuša [1], Peter Surový [2] [ID], Jiří Trombik [1] and Emanuel Kula [3]**

[1] Department of Forest Protection and Entomology, Czech University of Life Sciences Prague,
16500 Prague, Czech Republic; holusa@fld.czu.cz (J.H.); trombik@fld.czu.cz (J.T.)

[2] Department of Forest Management, Czech University of Life Sciences Prague, 16500 Prague, Czech Republic;
surovy@fld.czu.cz

[3] Department of Forest Protection and Wildlife Management, Mendel University in Brno,
61300 Brno, Czech Republic; kula@mendelu.cz

* Correspondence: resnerovak@fld.czu.cz

**Abstract:** *Ips cembrae* is the most important bark beetle pest of larches and has had several local outbreaks in recent decades in Europe. In this study, we compared the numbers of *I. cembrae* captured by pyramid-trap piles, trap trees, pheromone traps, and poisoned and baited tripods. We also studied how the properties of trap trees and trap logs (volume, sun exposure, and position relative to the ground once deployed) affected the trapping of *I. cembrae*. We found that both sexes avoided infestation at the bottom of the logs and more than 15 times the number of beetles were captured by traditional trap trees than by pheromone traps or baited and insecticide-treated tripods. The number of *I. cembrae* per trap tree did not decrease with trap volume; therefore, it is appropriate to use traps of small dimensions. Baited tripods, pyramid-trap piles, and pheromone traps could be useful for detection of the beginning of flight activity, but trap trees are the most useful for reducing *I. cembrae* numbers.

**Keywords:** large larch bark beetle; pheromone traps; trap trees; tripods; trap tree volume; insecticide

## 1. Introduction

European larch (*Larix decidua* Mill.) is native to the Alps and to several mountainous ranges in Central Europe. Its early introduction outside its native range, especially in the lowlands of western and northern Europe, has been problematic for ecological and phytosanitary reasons [1]. Outside of plantations, its current natural distribution is fragmented and spans about 500,000 ha. Beyond its native range, plantations of European larch cover an additional 500,000 ha [2]. In the Czech Republic, *L. decidua* covers a total area of 100,263 ha—i.e., 3.2% of the forested area in the country. Larch has been recently considered as useful for the regeneration of areas that have been cleared following bark beetle outbreaks in spruce stands [3].

*Ips cembrae* (Heer, 1836), one of several *Ips* species native to Europe, is an aggressive pest of larch, and local outbreaks of *I. cembrae* have been recorded in Europe [4–7]. In the years 2009–2019 in the Czech Republic, a total of 39,000 m$^3$ of larch was harvested, which is more than 1000 times less than the spruce harvested in the same period. This is the main reason why larch bark beetles are currently receiving less attention than spruce bark beetles by both researchers and foresters.

*Ips cembrae* attacks larches (*Larix* spp.) [6,8,9] of all ages without significant preference for altitude (400–2400 m) [10–12] It has no competitors for the use of the trunk and can multiply in monocultures

that have been established as substitute tree stands in areas with serious air pollution, but also on individual larches within forests dominated by other tree species [13]. The beetle will occasionally attack other conifers (*Picea* spp., *Abies* spp., or *Pinus* spp.) [10,14]. In addition to reproducing in weakened trees, *I. cembrae* also reproduces in healthy trees [6,15,16] and in felled wood [6]. Predicted changes in climate are likely to increase the effects of this species [17,18], especially outside the natural range of larch where outbreaks on stressed trees may occur [19–21]. Apart from the damage caused by the adult beetles, some ophiostomatoid fungi associated with *I. cembrae* are likely to harm the attacked trees [22–24].

The life cycle of *I. cembrae* can be considered to begin when males search for and penetrate suitable host trees. One to seven (most often three) females join each male in a nuptial chamber [9], and each female then creates a maternal gallery, which first radiates in a stellate pattern and then follows the fibers of the phloem [25]. Up to about 50 eggs are laid singly along each maternal gallery. The adults often emerge from their first gallery system and start a sister brood in another tree. Depending on altitude and climate, there is one generation or two overlapping generations per year. *Ips cembrae* can benefit from climatic warming because it can complete up to two generations in a hot growing season [7,16]. Before producing their own brood, the young adults must undergo maturation feeding, either within the phloem of the tree where they developed or in 2- to 18-year-old branches [25,26].

As with all bark beetles, the most effective control of *I. cembrae* is the consistent searching for and removal of infested trees (including branches) [4,27–29]. Monitoring of *I. cembrae* includes visual inspection and capture with pheromone traps, trap trees/logs, and poisoned traps [25]. In this study, we compared these methods of capturing *I. cembrae* with the goal of both monitoring and reducing its numbers.

A review of the literature indicated that little is known about the effectiveness of methods that could be used to capture *I. cembrae*. Based on the literature for other bark beetles, we assumed that the use of pheromone lures and trap trees would have similar abilities to reduce the population densities of *I. cembrae*. We also assumed that trap trees used to capture *I. cembrae* should be relatively large as is the case with *I. typographus* (Linnaeus, 1758) spruce trap trees—i.e., the *I. cembrae* larch trap trees should have a diameter at breast height of about 30 cm [30]. We also recognize that the ability of trap trees to capture the beetles and reduce their number could be small due to trap saturation or to the limited area for establishing galleries [31]. In contrast, pheromone traps and "tripod traps" (described in the next section) have a theoretically unlimited capacity to capture bark beetles [32] and thus to reduce *I. cembrae* abundance.

When attacking larch, *I. cembrae* establishes gallery systems from the bottom to the top of the tree [13]. Consequently, *I. cembrae* is able to develop on thin trunks, thick branches, and logging residues [33]. We therefore assessed how the ability of trap trees and trap logs to capture *I. cembrae* is influenced by their diameter and volume, exposure to sunlight, and position relative to the soil. This knowledge will help forest managers optimize defense measures.

The aims of this study were to compare *I. cembrae* capture methods and to find the best ways to install and use trap trees and trap logs against this pest. To accomplish our main aims above, we propose the following specific subaims: (i) to evaluate, in experiment 1, the number of *I. cembrae* captured that were affected by exposure of deployed trap logs to sunlight and the position of trap logs relative to their location relative to the soil surface once deployed; (ii) to compare, in experiment 2, the effects of larch logs arranged in standing pyramid-trap piles vs. trap trees laid on the ground on the number of *I. cembrae* captured; (iii) to determine, in experiment 3, whether the number of imagoes of *I. cembrae* captured on trap trees is affected by the volume of trap; to determine, in experiment 4, whether the type of the baited traps affects the numbers of *I. cembrae* beetles captured and whether there will be differences in the catches of both sexes; to evaluate, in experiment 5, and select the most effective conventional *I. cembrae* capture method applicable to forestry operations.

## 2. Materials and Methods

### 2.1. Study Sites

Experiments 1 and 2 were conducted on sites with monocultures of *Larix decidua*, and experiments 3–5 were conducted on sites that were mainly monocultures of spruce, *Picea abies* (L.) H. Karst., with a 5 to 10% mixture of larch trees that were more than 60 years old. Based on the willingness of forest owners to cooperate and older data on sites with high *I. cembrae* abundance in the last 15 years [34], we selected 11 study sites and one transect (experiment 2), which were located across the entire territory of the Czech Republic and at elevations ranging from 320 to 680 m a.s.l. (Figure 1). The selection of suitable study sites for the study of *I. cembrae* populations is complicated in the Czech Republic, in part because *I. cembrae* is considered less important than spruce bark beetles. As a consequence, the records of forest managers with respect to *I. cembrae* are seldom detailed, and *I. cembrae* abundance is very often underestimated, even though the quantity of the larch timber harvested has greatly increased in recent years [35].

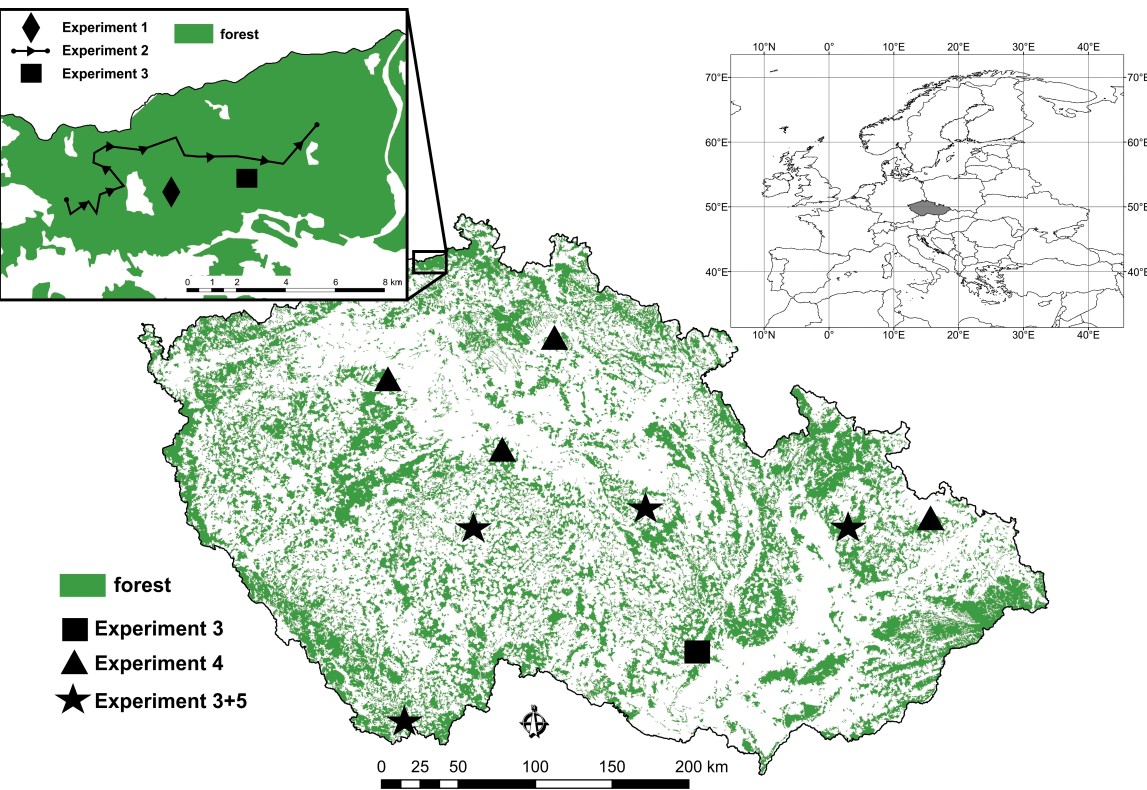

**Figure 1.** Study sites in the Czech Republic where experiments 1–5 were conducted from 2014 to 2019. As indicated, most experiments were conducted at more than one site, and two experiments were sometimes conducted at the same site.

### 2.2. Experiment 1: Influence of Sun Exposure and Position on Infestation of Trap Logs by I. cembrae

In February 2014, several healthy larch trees (*L. decidua*) were felled at the Sněžník study site (GPS: 50°47′43″ N, 14°05′34″ E; Figure 1) at an altitude of 600 m a.s.l. These trees were then cut into 150 cm-long logs that were leaning against a wooden railing (Figure 2). A total of 139 logs were installed in an east–west direction in an open area that was established by the harvesting of *Picea pungens* Engelm. and that was adjacent to an extensive larch monoculture. The bottoms of the logs were kept separate from the soil by a birch board. The volume, diameter in centimeters, and width of the phloem were measured for each log: averages (±SD) were 13.2 ± 10.0 dm³ for volume, 9.9 ± 3.7 cm for diameter, and 2.3 ± 0.7 mm for phloem width.

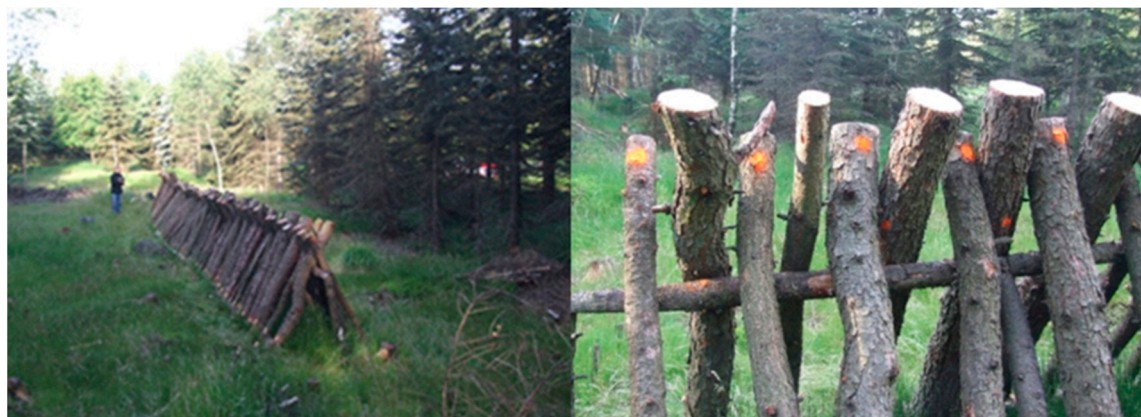

**Figure 2.** Design of experiment 1: larch logs installed at the Sněžník study site in 2014. Photograph by Kula.

In May 2014, after *I. cembrae* had infested the logs and developed galleries, the numbers of entry holes and maternal galleries were determined on two sides of each log (the sunny and shaded side) and at three positions on each side (top = 30 cm below the top of the log; middle = in the middle of the log; bottom = 30 cm above the bottom of the log). The area of each bark sample was 10 × 10 cm. As a consequence, six bark samples (2 sides × 3 positions per side) were assessed on each log.

The effects of log side (sunny or shaded) on the numbers of entry holes and maternal galleries were determined with a two-sample t-test in R version 4.0.2. A GLM model (also in R version 4.0.2) with family of binomial distributions was used to compare the effects of position (top, middle, or bottom of deployed trap logs relative to the soil surface), and log diameter on the number of *I. cembrae* males (i.e., the number of nuptial chambers) and the number of *I. cembrae* maternal galleries (number of females).

*2.3. Experiment 2: Numbers of I. cembrae Captured by Pyramid-Trap Piles vs. Trap Trees*

Experiment 2 was conducted in 2015–2019 near the town of Sněžník (GPS 50.4849 N, 14.0717 E; 10–14 plots) at altitudes between 500 and 600 m a.s.l. (Figure 1). Pairs of traps (each pair consisting of one pyramid-trap pile and one trap tree, which are described in the following paragraphs) were located along the edge of the larch stand but at least 20 m from the stand. A total of 59 pairs of pyramid-trap piles and trap trees were deployed during the research—i.e., 13, 11, 14, 11, and 10 pairs were deployed in 2015, 2016, 2017, 2018, and 2019, respectively. Pairs of traps were always prepared and deployed in the first half of March.

Before they were cut, the trees used to make trap trees had diameters of 17.7 ± 3.7 cm at a height of 1.3 m above the soil surface. Each trap tree was a healthy larch that was cut about 0.3 m above the soil; the trap tree consisted of the top portion that was left in place on the soil surface (Figure 3). Each trap tree was cut into 1.5 m-long sections that were left in place; the upper sunlit side and the lower shaded side of each section was marked, and three positions (upper, middle, or lower) on each side were also marked with the upper section always being the thinner section.

Non-baited pyramid-trap piles were prepared to match the trap trees—i.e., the trees used were from the same site and were of similar size and age (Figure 3). The trees that were used to construct the piles were felled at the study site, and the felled portions were cut into 1.5 m-long logs; the logs were 15.6 ± 2.9 cm thick. Each pyramid-trap pile consisted of 3–8 logs (depending on the height of the tree), which were arranged as shown in Figure 3.

As indicated, the trap tree and pyramid-trap piles were arranged in pairs with 10–15 m between each member of the pair and with 100–500 m between pairs. The volume of wood was 0.1 ± 0.1 m$^3$ per trap tree and 0.2 ± 0.1 m$^3$ per pyramid-trap pile.

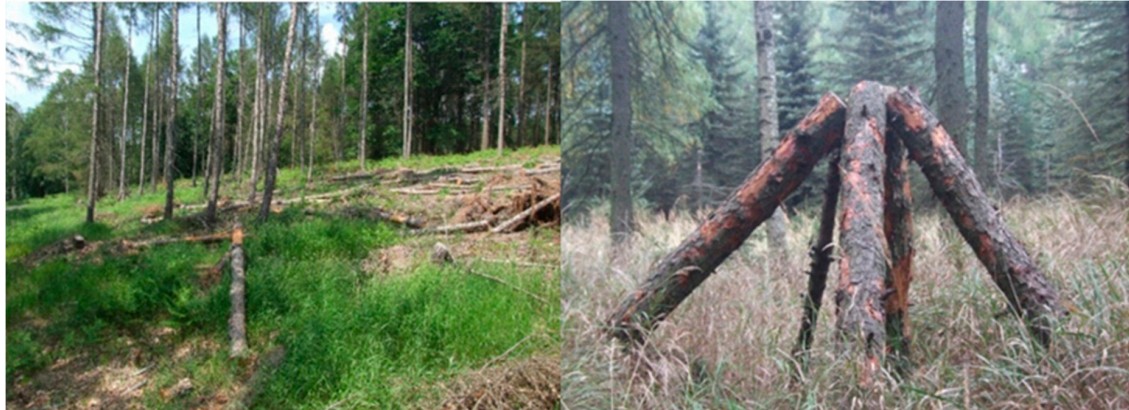

**Figure 3.** Larch trap tree (**left**) and pyramid-trap pile (**right**) and at the study site near Sněžník in 2019 (experiment 2). Photograph by Kula.

The traps were evaluated in May/June in the year of their deployment. Each trap tree was debarked at six sections (2 sides × 3 positions per side) and the number of *I. cembrae* in each section was determined. The pyramid-trap piles were evaluated in a similar manner—i.e., six sections were sampled on each of the logs in each pile (2 sides × 3 positions per side). For both kinds of traps, the number of *I. cembrae* was determined per section and was expressed as the number per dm$^2$.

The studied parameters were analyzed using the GLM model family of binomial distributions. Population densities on trap trees and pyramid-trap piles were compared using Wilcoxon matched pairs test. All analyses and figures were carried out in R version 4.0.2.

*2.4. Experiment 3: Number of I. cembrae Captured as Affected by Trap Tree Wood Volume*

A total of 62 trap trees were deployed at five study sites at altitudes ranging from 390 to 680 m a.s.l. during the years 2017–2019 (Table 1, Figure 1). Healthy larch trees were felled in mid-March. The trap trees were cut, debranched, and left at a distance of 20 m from the nearest trap and at least 20 m from the nearest stand. The traps trees were cut into 1.5 m-long sections.

**Table 1.** Study sites in years 2017–2019 in experiment 3.

| Study Site | Altitude (m a.s.l.) | GPS N | GPS E | Year | Number of Trap Trees |
|---|---|---|---|---|---|
| Dolní Babákov | 560 | 49.7985 | 15.8961 | 2017 | 4 |
| Húzová | 680 | 49.8189 | 17.3523 | 2017 | 3 |
| Jiřetice u Neustupova | 480 | 49.5917 | 14.7247 | 2017 | 10 |
| Kristin Hrádek | 510 | 50.8044 | 14.1301 | 2019 | 5 |
| Rožmberk | 680 | 48.6628 | 14.3776 | 2017 | 12 |
| Soběšice | 390 | 49.1536 | 16.3654 | 2019 | 28 |

The trees were evaluated in May/June—i.e., at the end of the *I. cembrae* infestation period at the larval stage of development. Four sections (sample areas) were designated on each trap tree according to the method of [36]. The first section (bottom) was located 0.5 to 1.0 m from the bottom of the tree; the second section (stem) was located midway between the bottom section and the beginning of the crown; the third section (middle) was located at the beginning of the crown; the fourth section (crown) was located in the center of the crown. Each trap tree was debarked, and the numbers of *I. cembrae* entry holes and maternal galleries were determined in each section. The total number of beetles on the entire surface of the trap tree was calculated based on the total numbers on the individual sections divided by the proportion of the total surface represented by the sum of the individual sections. The volume of wood in each trap tree was calculated according to the volume table of woody plants [37] using

data on the height of the tree and the diameter at a height of 1.3 m. The volume of wood per trap tree ranged from 0.14 to 1.40 m$^3$ with an average of 0.56 ± 0.29 m$^3$.

The relationships between the number of captured beetles and the volume of wood per trap were assessed with a Poisson Regression model in R version 4.0.2.

### 2.5. Experiment 4: Numbers of I. cembrae Beetles Captured by Baited Tripods vs. Pheromone Traps

Five pairs of tripods and pheromone traps were installed just before the expected start of *I. cembrae* flight activity during March and early April of 2016 and 2017 at each of four study sites (Table 2, Figure 4). The study sites were at altitudes ranging from 320 to 420 m a.s.l. The pairs of traps were left in the study sites throughout the flight activity periods, which usually ended late in August. The two traps composing each pair were 20 m apart, and the pairs were located 20 m from the forest stands and from other traps. Pheromone traps of the Theyson® type were placed at a height of 1.5 m from the ground on two wooden poles; a pheromone lure was hung inside the trap on a metal wire. A Cembräwit® pheromone lure (Witasek Pflanzenschutz GmbH, Vienna, Austria) was used for both types of traps and was replaced after 8 weeks.

**Table 2.** Study sites for experiment 4.

| Study Site | Altitude (m a.s.l.) | GPS N | GPS E | Year |
| --- | --- | --- | --- | --- |
| Nouzov | 405 | 50.2202 | 13.9445 | 2016 |
| Stříbrná Skalice | 420 | 49.9196 | 14.8432 | 2016 |
| Žehrov | 320 | 50.5224 | 15.0913 | 2016 |
| Raduň | 400 | 49.8809 | 17.9548 | 2017 |

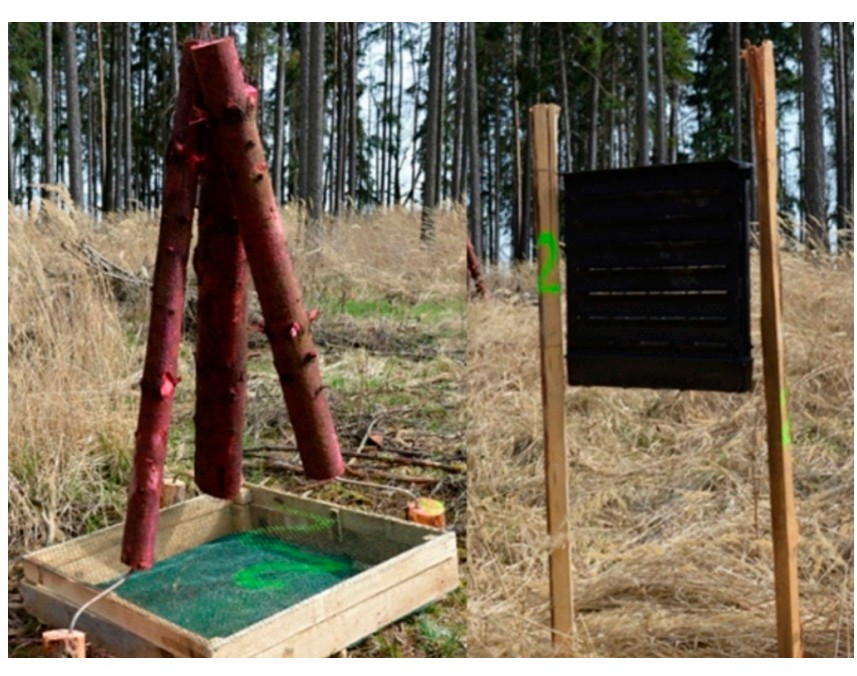

**Figure 4.** Baited tripod (**left**) and pheromone trap (**right**) at the Nouzov study site in 2017 (experiment 4). Photograph by Resnerová.

Tripod traps were constructed from felled healthy larch trees. For each tripod trap, three logs about 2 m long and 20 cm in diameter were cut and transported to the deployment location. The tops of the three logs were firmly connected to each other with a T-shaped piece of iron. After the iron was attached to the tops of the logs with nails, the logs were arranged in a tripod shape above two partially buried logs of non-target wood (spruce). Two 12 mm-diameter pieces of iron that extended from the spruce logs (one piece of iron per spruce log) were inserted into the bottoms of two of the three logs

that formed the larch tripod. A Cembräwit® pheromone lure was suspended at the top of each tripod so that it was at least partially protected from direct sunlight. The entire surface of each tripod was then treated with a mixture of Vaztak 10 SC® insecticide and a 1% solution of Scolycid® dye in water. The tripods were generally sprayed at 4- to 5-week intervals as needed, depending on the weather. Beetles that were attracted to the tripod and that died and fell were collected on a stainless-steel screen that was located beneath the tripod and that had narrow (<1 mm) openings. A second screen made of plastic and that had wide openings (15 mm) was positioned above the stainless-steel screen. The plastic screen allowed beetles to fall through but caught branches and other unwanted material and prevented animals from feeding on or otherwise removing the beetles on the stainless-steel screen.

*Ips cembrae* were collected every 7–10 days from the traps and tripods. In the laboratory, individuals of *I. cembrae* were identified with the aid of a stereomicroscope, and the sex was determined based on examination of genitals for at least 20 individuals per sample, datum, and site.

The total number of *I. cembrae* trapped and the proportions of males and females were compared for pheromone-baited tree traps vs. baited tripods with two-sample t-test in R version 4.0.2. We did not take into account the issue of different years though we do not expect differences.

### 2.6. Experiment 5: Number of I. cembrae Captured by Trap Trees, Baited Tripods, and Pheromone Traps

Experiment 5 was conducted in 2017 at four study sites at altitudes ranging from 480 to 680 m a.s.l. The experiment included 18 groups of traps with each group containing three types of traps (Table 3, Figure 1). The design of the experiment with respect to pheromone traps and baited tripods corresponded to that of experiment 4, but in addition to a pheromone trap and a tripod, each group included a trap tree.

**Table 3.** Study sites for experiment 5. Each group included three kinds of traps.

| Study Site | Altitude (m a.s.l.) | GPS N | GPS E | Year | Number of Groups |
|---|---|---|---|---|---|
| Dolní Babákov | 560 | 49.7985 | 15.8961 | 2017 | 4 |
| Húzová | 680 | 49.8189 | 17.3523 | 2017 | 3 |
| Jiřetice u Neustupova | 480 | 49.5917 | 14.7247 | 2017 | 5 |
| Rožmberk | 680 | 48.6628 | 14.3776 | 2017 | 6 |

The trap trees (mean wood volume 0.7 ± 0.4) were cut, debranched, and laid on the soil as described for experiment 3. Trap trees were cut and deployed in March. In May/June, the beetles captured by the trap trees were counted according to the methods described by [36] and for experiment 3. The following data were collected: the dimensions of the trap trees including the diameter at breast height (1.3 m from the base of the trunk), the number of entry holes and the number of maternal galleries on the debarked sections.

According to the phloem area of each trap tree and the number of entry holes (equivalent to the number of males) and maternal galleries (equivalent to the number of females), the total number of individuals in the entire trap tree was calculated.

Overall differences in beetle catches in the three types of traps were tested using Friedman's test (ANOVA). Poisson generalized linear mixed model (GLMM) with study site as a random effect was used to determine the relationships between the number of beetles captured and trap type. All analyses and figures were carried out in R version 4.0.2.

## 3. Results

### 3.1. Experiment 1: Influence of Sun Exposure and Position on Infestation of Trap Logs by I. cembrae

*Ips cembrae* penetrated logs with a diameter > 3.6 cm and a phloem width > 2 mm. The number of entry holes per $dm^2$ (mean ± SE) did not significantly differ between the sunny side (0.8 ± 0.6)

vs. the shady side (0.8 ± 0.5). The number of maternal galleries per $dm^2$ was higher on the sunny side (2.9 ± 1.7) than on the shady side (2.5 ± 1.6), but the difference was statistically insignificant according to the two-sample *t*-test ($t = 1.69$; $p < 0.10$).

The GLM model indicated that the number of *I. cembrae* males (i.e., the number of nuptial chambers) and females per $dm^2$ was positively related to log diameter (Tables 4 and 5). Both sexes avoided forming gallery systems at the bottom position of the deployed logs (i.e., the position closest to the ground); the latter position was included in the intercept of the model (Tables 4 and 5, Figure 5).

**Table 4.** Results of a GLM model describing the relationship between the number of entry holes of *Ips cembrae* per $dm^2$ and the following variables in experiment 1: position on the log (top, middle, and bottom = included in Intercept); diameter. Corrected Akaike's Information Criterion (AIC) = 210.09. Number of Fisher Scoring iterations was 5. ***, and * indicate significance at $p < 0.001$, and 0.05, respectively.

| Variable | Estimate | Std. Error | z Value | p Value |
|---|---|---|---|---|
| (Intercept) | −2.38 | 0.41306 | −5.758 | <0.001 *** |
| top | 0.31 | 0.31184 | 1.005 | 0.3150 |
| middle | 0.30 | 0.31228 | 0.968 | 0.3330 |
| diameter | 0.30 | 0.03239 | 2.279 | 0.0227 * |

**Table 5.** Results of a GLM model describing the relationship between the number of maternal galleries of *Ips cembrae* per $dm^2$ and the following variables in experiment 1: position on the log (top, middle, or bottom = included in Intercept); diameter. Corrected Akaike's Information Criterion (AIC) = 330.33. Number of Fisher Scoring iterations was 4. *** and ** indicate significance at $p < 0.001$ and 0.01, respectively.

| Variable | Estimate | Std. Error | z Value | p Value |
|---|---|---|---|---|
| (Intercept) | −2.01 | 0.36960 | −5.428 | <0.001 *** |
| top | 0.45 | 0.27417 | 1.657 | 0.09758 |
| middle | 0.14 | 0.28219 | 0.521 | 0.60220 |
| diameter | 0.08 | 0.02953 | 2.709 | 0.00675 ** |

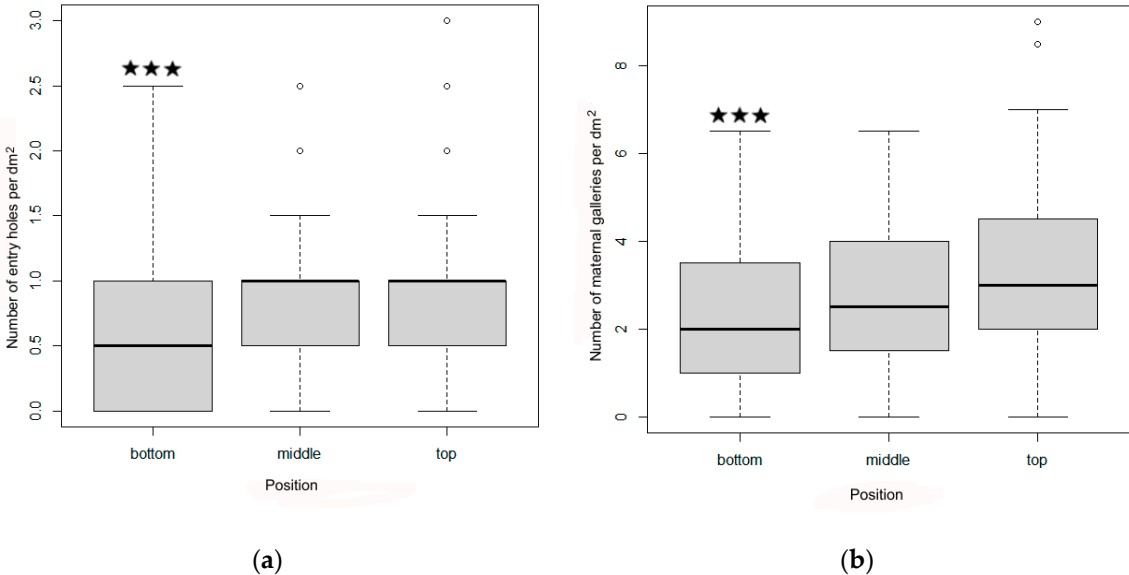

(**a**)          (**b**)

**Figure 5.** Number of entry holes (**a**) and maternal galleries (**b**) at top, middle, and bottom positions of the logs in experiment 1. The boxes indicate medians and interquartile range, the whiskers indicate minimum and maximum values, and circles indicate outliers. *** indicate significance at $p < 0.001$.

## 3.2. Experiment 2: The Abundance of I. cembrae in Unbaited Pyramid-Trap Piles vs. Trap Trees

The number of *I. cembrae* entry holes averaged $0.6 \pm 0.8$ per $dm^2$ in trap trees and $0.4 \pm 0.6$ per $dm^2$ in pyramid-trap piles. The numbers of entry holes were not significantly affected by trap type, diameter of wood, or volume of wood according to a Wilcoxon matched-pairs signed ranks test ($z = 1.65$, $p > 0.05$; Figure 6) and to a GLM analysis, which also showed no effect of year.

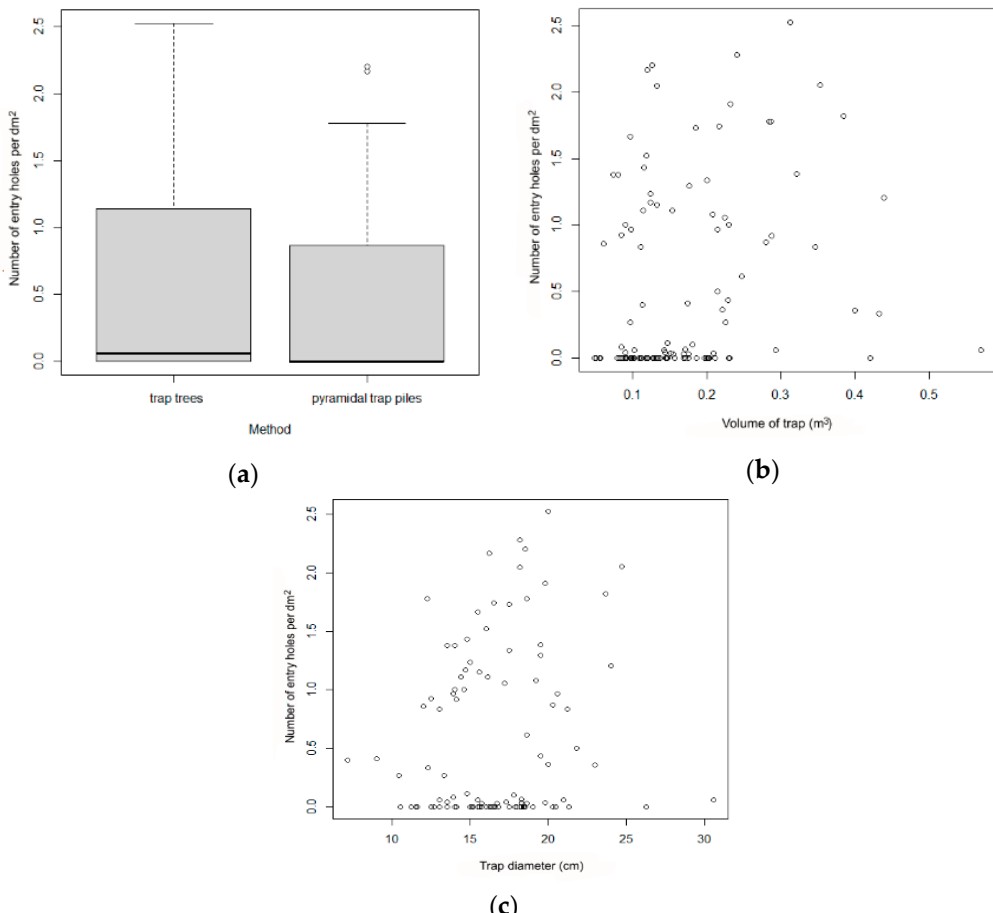

**Figure 6.** Number of entry holes as affected by trap type (trap trees vs. pyramid-trap piles) (**a**), wood volume (**b**), and wood diameter (**c**) in experiment 2. For trap type, the boxes indicate medians and interquartile range, the whiskers indicate minimum and maximum values, and circles indicate outliers.

## 3.3. Experiment 3: Number of I. cembrae Captured as Affected by Wood Volume of Trap Trees

In experiment 3, a total of 62 trap trees were analyzed at six study sites at altitudes ranging from 390 to 680 m a.s.l. A total of 257,010 *I. cembrae* were trapped, with a mean of 5039 ($\pm$5887) *I. cembrae* per trap tree.

The presented Poisson Regression model indicated that the number of *I. cembrae* per trap decreased with trap tree volume (Table 6, Figure 7).

**Table 6.** Results of a GLM model describing the relationship between the number of *Ips cembrae* per trap tree and the volume of wood per trap (experiment 3). Corrected Akaike's information criterion (AIC) = 312953. *** indicates significance at $p < 0.001$.

| Variable | Estimate | Std. Error | z Value | p Value |
|---|---|---|---|---|
| (Intercept) | 9.03 | 0.00398 | 2268.5 | <0.001 *** |
| volume | −1.11 | 0.00705 | −156.9 | <0.001 *** |

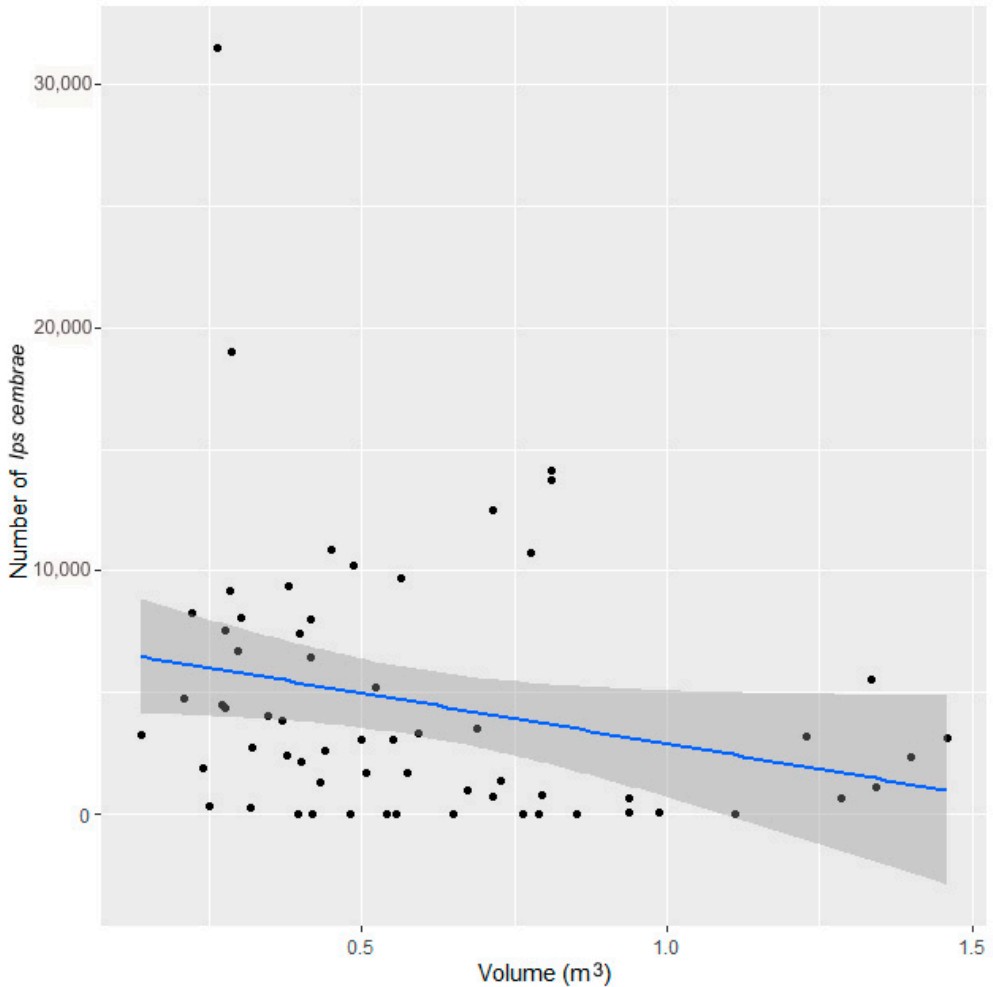

**Figure 7.** Relationships between the number of *I. cembrae* (points) per trap tree and wood volume per trap tree in experiment 3.

### 3.4. Experiment 4: Numbers of I. cembrae Beetles Captured by Baited Tripods vs. Pheromone Traps

A total of 13,617 *I. cembrae* were captured by baited tripods, and 10,789 were captured by pheromone traps. The sex ratio (males to females) was on average 1.5 for tripods and 0.9 for pheromone traps (Table 7). Although more individuals tended to be caught by tripods than by pheromone traps at all study sites (Table 7), the difference was not statistically significant (two-sample *t*-test: $t = 1.62$, $p > 0.05$).

**Table 7.** Numbers of *I. cembrae* captured by pheromone traps and tripods in experiment 4.

| Capture Methods and Variables/Study Site | S. Skalice | Raduň | Žehrov | Nouzov |
|---|---|---|---|---|
| Tripods—total number captured | 1655 | 3813 | 1200 | 6949 |
| Number of beetles identified as males | 490 | 483 | 606 | 666 |
| Number of beetles identified as females | 395 | 197 | 594 | 486 |
| Mean number ± SD per trap | 331 ± 269 | 763 ± 91 | 240 ± 44 | 1390 ± 916 |
| Sex ratio | 1.2 | 2.5 | 1.0 | 1.4 |
| Pheromone traps—total number captured | 340 | 3293 | 1038 | 6118 |
| Number of males | 144 | 356 | 353 | 437 |
| Number of females | 196 | 244 | 685 | 573 |
| Mean ± SD per trap | 68 ± 22 | 659 ± 135 | 208 ± 62 | 1224 ± 87 |
| Sex ratio | 0.7 | 1.5 | 0.5 | 0.8 |

Significantly more males were captured by pheromone-baited tripods than by pheromone traps (two-sample *t*-test: $t = 6.38$, $p < 0.00001$). The number of females captured did not significantly differ between the two kinds of traps (two-sample t-test: t = −0.13, $p > 0.05$) (Table 7, Figure 8).

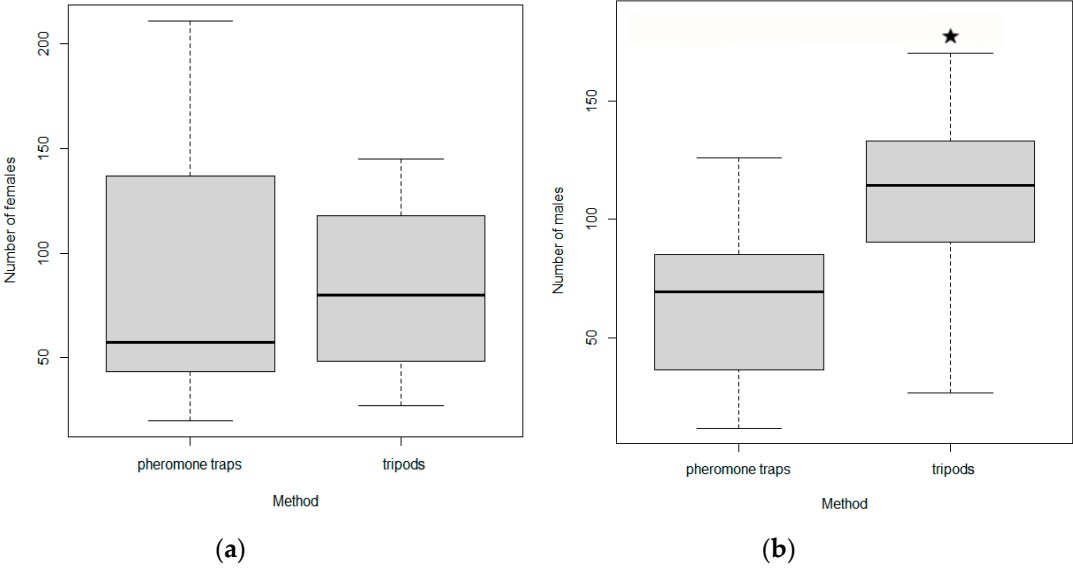

(**a**)                                                                (**b**)

**Figure 8.** Number *I. cembrae* females (**a**) and males (**b**) captured in pheromone traps and tripods in experiment 4. The boxes indicate medians and interquartile range, and the whiskers indicate minimum and maximum values. * indicate significance at $p < 0.05$.

*3.5. Experiment 5: Number of I. cembrae Captured by Trap Trees, Baited Tripods, and Pheromone Traps*

More *I. cembrae* were captured with trap trees than with baited tripods or pheromone traps except at the Dolní Babákov study site (Table 8, Figure 9). The total number of *I. cembrae* captured was 5432 for tripods, 5676 for pheromone traps, and 91,364 for trap trees. On average, trap trees captured 16 times more individuals than pheromone traps and 17 times more individuals than tripods during *I. cembrae* flight activity (Table 8). The number of trapped beetles was significantly higher with tree traps than with pheromone traps or tripods according to Friedman's test (ANOVA) ((N = 18, df = 2) = 8.33; $p < 0.05$). The difference between tripods and pheromone traps was minimal (Table 9).

**Table 8.** Numbers of *I. cembrae* captured in baited tripod traps, pheromone traps, and trap trees in experiment 5.

| Capture Methods and Variables/Study Site | Dolní Babákov | Húzová | Jiřetice u N. | Rožmberk |
|---|---|---|---|---|
| Tripods—total number captured | 1881 | 810 | 1029 | 1712 |
| Mean ± SD per trap | 479 ± 147 | 270 ± 24 | 205 ± 137 | 285 ± 196 |
| Pheromone traps—total number captured | 1819 | 1037 | 2348 | 472 |
| Mean ± SD per trap | 455 ± 79 | 346 ± 154 | 470 ± 361 | 79 ± 24 |
| Trap trees—total number captured | 8 | 58,555 | 17,260 | 15,541 |
| Mean ± SD per trap | 2 ± 2 | 19,518 ± 9588 | 3452 ± 3121 | 2590 ± 1669 |

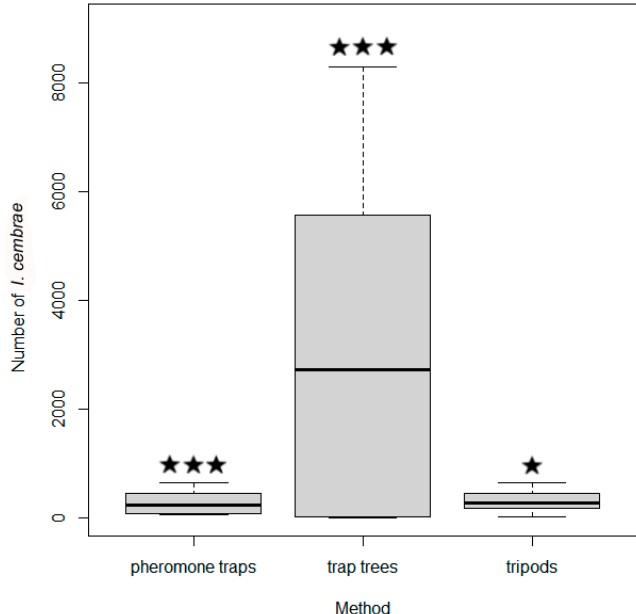

**Figure 9.** Numbers of *I. cembrae* captured by pheromone traps, trap trees, and tripods in experiment 5. The boxes indicate medians and interquartile range, and the whiskers indicate minimum and maximum values. *** and * indicate significance at $p < 0.001$ and 0.05, respectively.

**Table 9.** Results of a generalized linear mixed model (GLMM) describing the relationship between the number of *Ips cembrae* beetles per trap and trap type method and study site as random effect in experiment 5. Corrected Akaike's information criterion (AIC) = 67330. *** and * indicate significance at $p < 0.001$ and 0.05, respectively.

| Variable | Estimate | Std. Error | z Value | p Value |
|----------|----------|-----------|---------|---------|
| (Intercept) | 5.37 | 0.54996 | 9.767 | <0.001 *** |
| trap trees | 2.78 | 0.01368 | 203.126 | <0.001 *** |
| baited tripods | −0.04 | 0.01898 | −2.315 | 0.0206 * |

## 4. Discussion

In the current research, we compared the ability of several kinds of traps to capture *I. cembrae*. For traps that use wood, we also assessed the effects of the following variables on capture of *I. cembrae*: the volume of wood in the trap, the position on the wood relative to sun exposure, and the position on the wood relative to trunk height or the soil surface. In addition to assessing several methods that use pheromone lures, we assessed traditional trap trees and trap logs without pheromone lures.

Our results indicate that larch trap trees performed better than the other methods for the capture of *I. cembrae*, apparently because the beetle responds quickly to this resource and is attracted to wood regardless of its original location along the trunk. In contrast to other bark beetles, *I. cembrae* does not have a strong preference for a certain part of the tree, although some authors have noted a preference for relatively large-diameter trees [38]. As a control measure against *I. cembrae* outbreak, it is possible to choose trap trees of low volume within a given stand (experiment 3). This is inconsistent with the hypothesis that the effectiveness of spruce trap trees could depend on the size of the trees because tree size could determine the amount of host-specific substances released [39]. Unlike trap trees deployed against spruce bark beetles [30], larch trap trees deployed to control *I. cembrae* should not be covered with branches that lack needles in March.

Our results regarding the effects of sun exposure and position (experiment 1) provide strong evidence that *I. cembrae* attacks logs with a thickness > 3 cm, which confirms a similar study [13] and that *I. cembrae* may attack young larch stands between 2 and 18 years of age [7,25,40] and branches during

maturation feeding [25]. *I. cembrae* beetles probably fly at many heights above the soil, but in some cases, they apparently prefer to infest higher in the crown than in the lower sections [13]. The latter result was supported by the current results, which indicated a preference for higher positions on the traps logs. The results of experiment 1 indicate that both sexes showed no side (sunny vs. shady) preference when creating a gallery system on trap logs. One reason for the lack of preference by beetles might be that the phloem of larch trees exhibits similar defense reactions against *I. cembrae* regardless of the position along the trunk [41].

In the current research, we were interested in determining whether natural logs were more attractive to *I. cembrae* if they were positioned vertically rather than horizontally. For example, our comparison of trap trees and pyramid-trap piles indicated whether capture would be different when the same source material differed in arrangement. Trap trees should be more similar to the natural host resources used by bark beetles because they have a larger surface area than pyramid-trap pile. According to a previous study, trap logs, regardless of their arrangement, capture similar numbers of beetles whether they have or do not have a pheromone lure [42]. The latter study also found that, among trap log arrangements, crossed trap logs and pyramid-pile traps (as used in experiment 2) were the most effective at capturing *I. cembrae*. With pyramid-pile traps, *I. cembrae* produced substantially more entry holes per $dm^2$ in the study by [42] than in the current study. Our results (experiment 2) confirm that the addition of pheromone lures to trap logs, pyramid-trap piles, or trap trees is useless, because fresh wood is sufficiently attractive to *I. cembrae* (see also [42]). Baited and unbaited traps composed of larch logs have similar attractiveness because the traps look like a tree or its parts, or because they release attractive volatiles. The high number of beetles captured in the baited traps in experiments 4 and 5 confirm that pheromone lures attract *I. cembrae* but are more useful for monitoring *I. cembrae* in the forest (i.e., for determining the timing and peak of flight activity) than for defending against infestation [42]. A potential disadvantage of using pheromone traps for mass trapping is that they may attract beetles from long distances and thereby increase *I. cembrae* numbers in areas where the beetles' abundance was previously low, as has been reported for other bark beetle species [43]. Even though pheromone-baited traps captured large numbers of beetles, enough beetles would remain untrapped to colonize susceptible hosts, quickly replenishing the population density due to low intraspecific competition. In agreement with other researchers [44,45], we recommend using pheromone traps for detection in locations where the occurrence of *I. cembrae* has not been confirmed.

Both males and females respond to *I. cembrae* aggregation pheromones [46,47]. In the current study, the female:male sex ratio in pheromone traps fluctuated between 1 and 2, but the frequency of female and male beetles in traps was not significantly different [34]. *I. cembrae* was attracted more to tripods than to pheromone traps in experiment 4 of the current study, which differs from the results previously obtained for *I. duplicatus* (Sahlberg, 1836) and *I. typographus* [48,49]. The significantly greater number of males captured in tripods than in pheromone traps in experiment 4 was consistent with previous reports for other species of bark beetles in the genus *Ips* [48–50]. Males or females establishing gallery systems of bark beetles and ambrosia beetles are usually attracted in greater numbers to traps with visual stimuli that resemble tree trunk silhouettes than to traps baited only with host odors or pheromones [43,51–54].

Pheromone-baited tripod traps that have been sprayed with insecticide are effective against *I. cembrae* because such traps attract males, and it is males that establish the gallery systems. The possibility of underestimating the capture of *I. cembrae* on poisoned tripods because of a sublethal effect has been ruled out by a number of laboratory and field observations that used lambda-cyhalothrin [55] or alpha-cypermethrin as an insecticide [56,57], as was the case in our study.

The use of baited tripods is thus more effective than pheromone traps in monitoring *I. cembrae*, but due to the complexity of baited tripod installation, it is up to the forest managers to decide which trap to use. On the other hand, among the best defenses in addition to the timely detection and removal of bark beetle-infested trees is the use of trap trees. In studies comparing trap trees and baited tripods

for monitoring *I. typographus*, trap trees were found to be more effective for all generations [58] or at least for the overwintering generation of *I. typographus* [56].

A previous study found that the area of the trapping surface of commercial traps was insufficient for optimal trapping efficiency [59]. Because a trap tree has a larger trapping surface than a commercial trap, trapping efficiency should be greater with trap trees than with commercial traps. When the population density of *I. cembrae* is low, however, the aggregation pheromones seem to be especially effective in attracting the beetles that initially fly into a site, such that baited and poisoned tripods and pheromone traps may be more effective than trap trees early in the season, as appeared to be the case at the Dolní Babákov study site in experiment 5 (Table 8).

The results of experiment 5 showed that, except at the the Dolní Babákov study site (where the population density of *I. cembrae* was probably low), trap trees captured at least 15 times more beetles than pheromone traps or baited poisoned tripods. This difference in efficacy was even greater than that previously reported for comparisons of the trapping of spruce bark beetles by spruce trap trees and artificial traps [31,60]. Similarly, standing spruce trap trees baited with pheromones and treated with insecticides caught up to 30 times more beetles than pheromone traps [61]. In contrast to our results with baited tripods vs. pheromone traps, the number of *I. cembrae* captured did not significantly differ between standing trap logs (1–3 m long) vs. pheromone traps in a previous study [32]. The use of short logs rather than complete trees seems to greatly reduce the treated bark surface and consequently the natural attractiveness. Because the amount of host volatiles decreases with the surface area of the trap trees, the surface area of trap trees greatly influences their effectiveness [32].

## 5. Conclusions

As is the case for control of other bark beetles, control of *I. cembrae* requires the ongoing and timely search for infested trees or logging residues and their timely elimination, or the processing of all materials suitable for bark beetle reproduction. Traditional trap trees of relatively small size are the most effective tool for *I. cembrae* capture. Unbaited tripods or pyramid-trap piles can be used, but they are less effective than trap trees. Unbaited tripods and pyramid-trap piles should be placed in sunlit locations near the forest edge and should be raised above the ground (to increase the area suitable for infestation). Baited tripods and pheromone traps are suitable for monitoring *I. cembrae* in stands, ports, or landfills with imported timber, with tripods being more efficient because they capture more males than pheromone traps. The pheromone trapping is effective and useful, especially in areas with a high proportion of larch, where early detection of *I. cembrae* flight activity is required. However, the detection of *I. cembrae* presence with pheromone traps is unnecessary if the beetle is already known to be present in a stand. Similarly, the detection of *I. cembrae* flight activity with pheromone traps is unnecessary because *I. cembrae* does not have clear peaks of flight activity—i.e., its adults fly continuously during the growing season. If *I. cembrae* is known to be present in a stand, all larch material that is infested or that is likely to be infested should be removed and processed.

**Author Contributions:** K.R., J.H., and E.K. provided methodology of research, collected field and laboratory data, supervised the research, and edited the manuscript; P.S. and J.T. performed the statistical analyses and the visualizations. All authors have read and agreed to the published version of the manuscript.

**Funding:** This research was funded by the Ministry of Agriculture of the Czech Republic, grant number QK1920433.

**Acknowledgments:** The authors thank Bruce Jaffee (USA) for linguistic and editorial improvements, and Jakub Zounar, Nikola Bohatá, Jiří Doskočil, Oldřich Housa, Jiří Bidmon, Radka Melecká, Richard Hurný, and Markéta Vand'urková for help with the field work.

**Conflicts of Interest:** The authors declare no conflict of interest.

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
