# Peer review of "Comparison of Ips cembrae (Coleoptera: Curculionidae) Capture Methods: Small Trap Trees Caught the Most Beetles"

_forests, doi:10.3390/f11121275_

Round 1

Reviewer 1 Report

Thank you for the opportunity to review this manuscript, describing the various methods used to trap Ips cembrae and comparing between these methods. These types of studies are important, especially as they can impact on the operational management of these pest species and have major cost and efficiency implications for forest managers. I am concerned, however, with some of the statistical analyses, adjusting these may not change the results, but it appears that some of the analyses are using incorrect models or that there is no apparent check for violation of assumptions for some of the tests. I recommend reading

Zuur et al. 2010. A protocol for data exploration to avoid common statistical problems. Methods in Ecology and Evolution. 1: 3-14

I am concerned that some of the recommendations you make regarding on-going monitoring and control of this beetle may not be justified from the data presented

Minor points

Ln 25     replace “is among other Ips species” with “is one of several Ips species”

Ln 26     replace “aggressiveness” with “aggressive”

Ln 55-56   replace the three uses of “on” with “in”

Ln 61     replace “then begins create a maternal” with “then creates a maternal”

Ln 63     replace “systems” with “system” and “on” with “in”

Ln 64     replace “there are one or two overlapping generations per year” with “there is one generation, or two overlapping generations, per year”

Ln 70     remove the word “but”

Ln 101, 118, 140, 173, 192, 228 all the sub-sections of the Materials and Methods section start with a number 3 rather than a number 2, e.g. “3.1 Study sites” should be “2.1 Study sites”

Ln 107   it is hard to see that this is 12 sites, I assume from counting them up that you have counted the wiggly transect for Experiment 2 all as one site, but it is hard to justify this as one site, when all the others are much more single points

Ln 133   Did you check that your samples did not violate the assumptions of your statistical tests (e.g. independence, homogeneity of variance etc)? If you did then you should quote these tests (or describe that the issue was checked graphically), to justify parametric as opposed to non-parametric analyses. I note, for instance that you use the Welch’s two sample t-test, an appropriate test for unequal variances, but you don’t say why you used it, this is presumably because you did check, but you do not state this

Ln 133-4    You should examine whether or not there is co-linearity between the variables, especially volume and diameter – these are clearly not independent of each other as the length of the log was fixed at 1.5 m and by definition there is a relationship between diameter and volume. Why have you used a binomial distribution for this analysis – as count data this should be fitted to a Poisson distribution? Binomial distribution is used for binomial or proportional data

Ln 165   here you say that there were five logs in a pile, but above (ln 157) you say that the number of logs was between 3 and 8, which figure is correct?

Ln 176   why do you state here that the trap trees were treated the same as experiment 5, as the first time the method is discussed, it should be described here, and the experiment 5 can refer back to experiment 3

Ln 194   replace “at a total of four study sites” with “at each of four study sites”, otherwise to can be misconstrued that you had only five pairs in total, rather than twenty pairs of replicates.

Ln 195   replace “were at the altitudes” with “were at altitudes”

Ln 223   I am concerned that three of the sites were used in one year, and the fourth was the following year, does the inter-year differences affect the results? This issue should be addressed, at least, even if it had no effect

Ln 241   what do you mean by “bast area”, is this supposed to be “basal area”, or is this a reference to the term “bast” that is somewhat synonymous with “phloem”? I have never heard this term used before in English

Ln 256   why is the data on number of males between sunny and shady sides not shown? How is this different from the number of entry holes, is that not the proxy you are using for number of males?

Ln 258   replace “(2.5 ± 1.6), and the difference was statistically significant according to Welch’s two-sample t-test: t = 1.69 p <0.1).” with “(2.5 ± 1.6) (Welch’s two-sample t-test: t = 1.69 p <0.1).”

Ln 258   are these results actually different? As written your p-value is not less than 0.05, the normal value ascribed to alpha. Is this a typographical error? Have you left out a zero? If you are using a different value of alpha, then you need to justify this. Regardless given the variance shown in the SE I am surprised that these data are different.

Ln 263   you have no evidence to support your claim about female preference with respect to position on the log, there is no statistical preference for the top of the log, based on Fig 5 I would be more likely to agree that males show an avoidance for the bottom of the log, but the z-value for this variable is not given in Table 4 or 5, so I cannot be sure. Assuming that the value listed for “intercept” is that for the bottom of the log, then perhaps you do have evidence for a positional preference, but you do not state this anywhere

Table 4, 5, 6, 7, 10   In addition to the z and p-values please present the model estimate and standard error,  I also think that some form of post-hoc analysis is required here, to ascertain if there are differences between the different positions, rather than just indicating if each of the position was a significant contributor to the model  

Fig 5      I do not really understand your box and whisker plots, the normal things displayed in such a plot is the median and the interquartile range by the box, rather than the standard deviation, as you have stated. Is this really the standard deviation? Although the data will have a standard deviation, this is a measure of variation around the mean, not the median

Ln 281   I am concerned that the value of the standard errors shown here for the entry holes has a greater magnitude than mean number of holes, such that as expressed the range actually covers a negative number of holes. Is this correct?

Ln 288   Again, surely this is the interquartile range rather than the standard deviation

Ln 299   you have now used a Poisson regression model rather than binomial. I agree that this is appropriate for count data, by why did you previously use binomial?

Ln 300-1    I do not agree that the plot shows even a trend towards a relationship between number of beetles and altitude, there may be some relationship between altitude and volume (although I am not certain about that), but there are plenty of low altitude trees with similar numbers of beetles to the higher altitude ones.

Ln 354   replace “of following variables” with “of the following variables”

Ln 364-5    You cannot make this statement about position and side of the logs, there was no significant preference for side by males, and the value for females is questionable, and you present no data to support the preference for higher positions

Author Response

Reviewer 1

Authors: Thank you for your review and comments, we we tried to adjust and recalculate the models and adjust some statistical analyzes to be correct.

I am concerned that some of the recommendations you make regarding on-going monitoring and control of this beetle may not be justified from the data presented

 Authors: thank you for your comments, we understand that and restructured discussion of manuscript – and shortened it. We admit that this help improve manuscript a lot.

Minor points

Ln 25 replace “is among other Ips species” with “is one of several Ips species”; Ln 26 replace “aggressiveness” with “aggressive”; Ln 55-56 replace the three uses of “on” with “in”; Ln 61 replace “then begins create a maternal” with “then creates a maternal”; Ln 63 replace “systems” with “system” and “on” with “in”; Ln 64 replace “there are one or two overlapping generations per year” with “there is one generation, or two overlapping generations, per year”; Ln 70 remove the word “but”

Authors: all corrected or deleted when we restructured Introduction

Ln 101, 118, 140, 173, 192, 228 all the sub-sections of the Materials and Methods section start with a number 3 rather than a number 2, e.g. “3.1 Study sites” should be “2.1 Study sites”

Authors: corrected

Ln 107   it is hard to see that this is 12 sites, I assume from counting them up that you have counted the wiggly transect for Experiment 2 all as one site, but it is hard to justify this as one site, when all the others are much more single points

Authors: we understand this comment and change subscription in text: „Based on willingness of forest owners and older data on sites with high I. cembrae abundance in the last 15 years [34], we selected 11 study sites and one transect (experiment 2), which were located across the entire territory of the Czech Republic and at elevations ranging from 320 to 680 m a.s.l. (Figure 1).“

Ln 133   Did you check that your samples did not violate the assumptions of your statistical tests (e.g. independence, homogeneity of variance etc)? If you did then you should quote these tests (or describe that the issue was checked graphically), to justify parametric as opposed to non-parametric analyses. I note, for instance that you use the Welch’s two sample t-test, an appropriate test for unequal variances, but you don’t say why you used it, this is presumably because you did check, but you do not state this

Authors: we recalculated the data by two sample t-test for dependent samples, the data have a normal distribution

Ln 133-4    You should examine whether or not there is co-linearity between the variables, especially volume and diameter – these are clearly not independent of each other as the length of the log was fixed at 1.5 m and by definition there is a relationship between diameter and volume.

Authors: yes, this problem happened for some missunderstandment in data Exchange we removed in all the models the volume variable. Though most of the model variables did follow the previous behaviour

Why have you used a binomial distribution for this analysis – as count data this should be fitted to a Poisson distribution? Binomial distribution is used for binomial or proportional data

Authors: Yes, other reviewers also mentioned this, the reason for binomial was that the data was delivered not in counts of beetles, but in densities per dm2 in Experiment 1 and 2, we opted for binomial distribution for that purpose. 

Ln 165   here you say that there were five logs in a pile, but above (ln 157) you say that the number of logs was between 3 and 8, which figure is correct?

Authors: number of logs varied from 3 to 8 logs, corrected

Ln 176   why do you state here that the trap trees were treated the same as experiment 5, as the first time the method is discussed, it should be described here, and the experiment 5 can refer back to experiment 3

Authors: changed and part of methods moved from Exp 5 to Exp 3

Ln 194   replace “at a total of four study sites” with “at each of four study sites”, otherwise to can be misconstrued that you had only five pairs in total, rather than twenty pairs of replicates.

Authors: replaced

Ln 195   replace “were at the altitudes” with “were at altitudes”

Authors: replaced

Ln 223   I am concerned that three of the sites were used in one year, and the fourth was the following year, does the inter-year differences affect the results? This issue should be addressed, at least, even if it had no effect

Authors: We did not take to account the issue of different years though we do not expect differences. Added to methods.

Ln 241   what do you mean by “bast area”, is this supposed to be “basal area”, or is this a reference to the term “bast” that is somewhat synonymous with “phloem”? I have never heard this term used before in English

Authors: we mean phloem, corrected

Ln 256   why is the data on number of males between sunny and shady sides not shown? How is this different from the number of entry holes, is that not the proxy you are using for number of males?

Authors: deleted, it is the same (entry holes=number of males)

Ln 258   replace “(2.5 ± 1.6), and the difference was statistically significant according to Welch’s two-sample t-test: t = 1.69 p <0.1).” with “(2.5 ± 1.6) (Welch’s two-sample t-test: t = 1.69 p <0.1).”

Authors: changed to „The number of maternal galleries per dm2 was higher on the sunny side (2.9 ± 1.7) than on shady side (2.5 ± 1.6), but the difference was statistically insignificant according to two-sample t-test (t = 1.69; p <0.10).“

Ln 258   are these results actually different? As written your p-value is not less than 0.05, the normal value ascribed to alpha. Is this a typographical error? Have you left out a zero? If you are using a different value of alpha, then you need to justify this. Regardless given the variance shown in the SE I am surprised that these data are different.

Authors: yo are right, changed to „The number of maternal galleries per dm2 was higher on the sunny side (2.9 ± 1.7) than on shady side (2.5 ± 1.6), but the difference was statistically insignificant according to two-sample t-test (t = 1.69; p <0.10).“

Ln 263   you have no evidence to support your claim about female preference with respect to position on the log, there is no statistical preference for the top of the log, based on Fig 5 I would be more likely to agree that males show an avoidance for the bottom of the log, but the z-value for this variable is not given in Table 4 or 5, so I cannot be sure. Assuming that the value listed for “intercept” is that for the bottom of the log, then perhaps you do have evidence for a positional preference, but you do not state this anywhere

Authors: the intercept includes the bottom position we also tested the difference using anova the upper and middle do not differ among them

Table 4, 5, 6, 7, 10   In addition to the z and p-values please present the model estimate and standard error. I also think that some form of post-hoc analysis is required here, to ascertain if there are differences between the different positions, rather than just indicating if each of the position was a significant contributor to the model 

Authors: values are now presented. We described that the intercept includes the value of the bottom position this was also tested in other analysis, removing the intercept would cause that all three positions appear to contribute significantly to the model 

Fig 5      I do not really understand your box and whisker plots, the normal things displayed in such a plot is the median and the interquartile range by the box, rather than the standard deviation, as you have stated. Is this really the standard deviation? Although the data will have a standard deviation, this is a measure of variation around the mean, not the median

Authors: thank you for this comment, this is mistake, in the plots are median and interquartile range

Ln 281   I am concerned that the value of the standard errors shown here for the entry holes has a greater magnitude than mean number of holes, such that as expressed the range actually covers a negative number of holes. Is this correct?

Authors: We understand your comment, the data do not have a Gaussian distribution, and in this case the median should be used rather than the average, because the range does not cover negative values. However, the median would be 0 because many sections were unoccupied.

Ln 288   Again, surely this is the interquartile range rather than the standard deviation

Authors: thank you for comment, in the plots are median and interquartile range

Ln 299   you have now used a Poisson regression model rather than binomial. I agree that this is appropriate for count data, by why did you previously use binomial?

Authors: Yes, other reviewers also mentioned this, the reason for binomial was that the data was delivered not in counts of beetles, but in densities per dm2 in Experiment 1 and 2, we opted for binomial distribution for that purpose. Other experiments included count data and therefore we used Poisson regression model

Ln 300-1    I do not agree that the plot shows even a trend towards a relationship between number of beetles and altitude, there may be some relationship between altitude and volume (although I am not certain about that), but there are plenty of low altitude trees with similar numbers of beetles to the higher altitude ones.

Authors: Yes, if we wanted to analyze this topic better, more robust data would be needed. Therefore, according to other reviwers, we prefer to omit the altitude analysis from the manuscript and reduce the experiment.

Ln 354   replace “of following variables” with “of the following variables”

Authors: replaced

Ln 364-5    You cannot make this statement about position and side of the logs, there was no significant preference for side by males, and the value for females is questionable, and you present no data to support the preference for higher positions

Authors: you are right, we changed all statements about side preference

Reviewer 2 Report

This paper is a useful addition to the literature on controlling Ips cembrae for which there is a lack of information beyond pheromones traps. My comments (in the attached document) are largely minor but should help particularly for the general reader who is less informed about bark beetles and their control. 

The introduction is somewhat confusing, jumping back and forth between topics. Better organisation of the introduction to cover the topics in question would improve readability, e.g. impact of the pest on trees and economically, feeding preferences and mode of feeding, outbreaks, susceptibility of trees, current management strategies/gaps and need for this study.  

Author Response

Reviewer 2

Authors: Thank you for your very helpful comments and remarks on the unclear parts of our work. We tried to improve the manuscript according to all recommendations.

The introduction is somewhat confusing, jumping back and forth between topics. Better organisation of the introduction to cover the topics in question would improve readability, e.g. impact of the pest on trees and economically, feeding preferences and mode of feeding, outbreaks, susceptibility of trees, current management strategies/gaps and need for this study.  

Authors: thank you fot this comment, we understand that and restructured this part of manuscript – and shortened it. We admit that this help improve manuscript a lot. Due to the modification of the text, some of the commented parts in the introduction have been deleted.

PDF comments

Title: I was confused by this term until I read the methods and initially read it as a grammatical error. Is this a conventional 'trapping method' that others will understand? Is there better terminology that could be used? Why is not just a single log? Authors: with all due respect it is a conventional 'trapping method' for bark beetle trapping. Let us keep the title in its original version

Authors: with all due respect it is a conventional 'trapping method' for bark beetle trapping. Let us keep the title in its original version, e.g. Holuša, J.; Hlásny, T.; Modlinger, R.; Lukášová, K.; Kula, E. Felled trap trees as the traditional method for bark beetle control: Can the trapping performance be increased? Forest Ecology and Management 2017, 404, 165–173, doi:10.1016/j.foreco.2017.08.019.

Ln 14: t would be useful to add a timeframe, e.g. in the last few years, decades..etc.

Authors: added

Ln 25: several rather than other?

Authors: introduction is changed

Ln 26: aggressive rather than agressiveness, though I wonder if you actually mean destructive? Authors: changed

Ln 27 Inconsistencies in use of I. or Ips at the start of sentences.

Authors: fixed in manuscript

Ln 28: I am not sure what this means. It has no competitors for the use of the trunk?

Authors: indeed, reworded

Ln 31: This sentence seems out of place, either need to add detail as to consequences of attack or move further on in introduction when describing impact of beetle on trees. Instead add something on reasons why it is important to consider this pest in more detail or why it has not been so thus far (see comment further on in introduction).

Authors: we restructured this part

Ln 32: Previous records about the occurrence....

Authors: changed

Ln 38: Repetition of first paragraph, I would consolidate.

Authors: We changed organisation of the introduction to improve readability

Ln 39: Without significant preference for what? You have already mentioned age classes and altitude, is there another factor? Or reword to say without significant preference for age class or altitude.... Authors: We changed organisation of the introduction

Ln 44: Again, this might be best aluded to in the first paragraph to say that climatic predictions are likely to increase the impact of the pest.

Authors: We changed organisation of the introduction

Ln 70: Are any economic impacts known? I would move this to first paragraph. But also need to justify, if larch is a minor crop, is this pest really a problem warrenting further investigation? Perhaps part of that lies in its increased impact under climate change, the potential for it to attack other tree species, the potential for the species to increase its range and/or an increase in larch planting? Authors: restructured and aditional information added about increased impact of larch in future. See Introduction

Ln 89: This needs to come before you set out your plans for assessment as in previous paragraph. Authors: restructured

Ln 102: I would start with which sites you selected and then explain why these were based on older records.

Authors: we agree - restructured

Figure 1: Another symbol would be clearer for viewing at print scale, e.g. triangle or star.

Authors: we changed the symbols to stars.

Ln 156: remove dash between 1.5 and m

Authors: removed

Ln 168: I wonder if it's worth circling the 'trap tree' to make it clear to the reader?

Authors: trap tree is a traditional term, we will keep, but we have changed the figure label to be more clear

Ln 175: Typical of trees that are attacked? What properties were used to define that?

Authors: deleted

Ln 195: remove 'the'

Authors: deleted

Ln 239: families of what?

Authors: mistake in translation, deleted

Ln 241 I don't know what 'bast area' means.

Authors: changed to phloem area

Ln 243-245 These last two sentences seem superfluous.

Authors: deleted

Ln 258: add opening bracket or remove this one

Authors: changed

glm usually written as GLM?

Authors: changed

Ln 260 I am not sure I understand how this can be the case when the logs were all the same length, surely diameter would be positively related to volume and therefore the relationship with the beetles would be the same?

Authors: yes, this problem happened for some missunderstandment in data Exchange we removed in all the models the volume variable. Though most of the model variables did follow the previous behaviour

Ln 263 but this was not significant?

Authors, yes it is significant, bottom is in glm included in intercept, see bellow

Table 4 and 5: bottom is missing?

Authors: bottom is incuded in Intercept, our mistake, added to table headings and text

Ln 312 on average, or averaged

Authors: changed

Ln 323 pheromone-baited tripods

Authors: changed

Tables It would be more conventional to write <0.001

Authors: corrected in Tables

Ln 358 there is a mixture of tenses in this section

Authors: changed

Ln 370 provide strong evidence for, rather than prove?

Authors: changed

Ln 401 Perhaps move the information about pheromone compounds to introduction? It is information rather than a discursive point.

Authors: this part was deleted

Ln 415 This could be better worded. Peaks not detectable because they don't exist, but then later in paragraph you say there might be a peak in July.

Authors: changed and better worded

Ln 452 I think it is important to discuss why you think the trap trees did not work well at this site. Although the results at the other sites were excellent, it shows trap trees will not always work or will not work in certain circumstances and for the practioner that might be important information from which to judge which method to use.

Authors: added certain circumstances

Possible missing references:

Niemeyer H, 1989. First results with a pheromone-trap system for monitoring bark beetles in Lower Saxony and Schleswig-Holstein. Forst und Holz, 44(5):114-115.

Pavlin R, 1997. New locations of the larch bark beetle (Ips cembrae) in Slovenia. Gozdarski Vestnik, 55(7/8):336-342; 12 ref

Authors: references were added

Reviewer 3 Report

review "Comparison of Ips cembrae (Coleoptera: Curculionidae) capture methods: small trap trees caught the most beetles (forests-979093)"

I went through the manuscript "Comparison of Ips cembrae (Coleoptera: Curculionidae) capture methods: small trap trees caught the most beetles (forests-979093)". The authors made a large study including different experiments on the monitoring methods for Ips cembrae. In total 5 experiments were done in which first the position of infestation on the trap log were determined, second the way of setting the trap trees are situated, position of the location of the trap log, and different methods attracting the beetles. In short, the study shows that the Ips cembrae doesn’t differentiate among parts of the tree stem but is affected by the diameter and/or the volume. Also, there was no difference between unbaited pyramid-trap piles vs. trap trees. Furthermore, the trap trees are most effective catching method followed by baited tripods. Pheromone traps are least effective.

The manuscript is very confusingly written, and I do not know if this is due to the writing style or the setup of the experiments, but probably both.

First regarding the writing style. The introduction is very long and does not seem to have a consistent structure. Furthermore, the authors want to show that Ips cembrae is potentially important pest, however, mentioning all the time that there are only local outbreaks which it not necessary kill the trees. Although I think it is important that a study for this species is important, I think the economic importance should be put in perspective. My suggestion is that the introduction, but also the discussion should be shortened and restructured. It would be very helpful that the experts show per experiment still the sub aim. I also suggest that the discussion is divided into a part where the results and the management implications are discussed. Below I will be some additional minor comments.

Then regarding the set up, I have the feeling that the authors did several independent experiments and now want to combine it in one manuscript, because not all results showed promising results to be a stand alone paper. First, many of the results of the experiments are overlapping. In most of the experiments volume and diameter are considered, although it was only explicitly tested for experiment 3. Furthermore, in experiment 3 also the altitude was included, but only 5 locations were sampled. To get a good result for altitude more locations should be considered, because this result could also because of other different factors in the separate locations. No good conclusions can be made regarding the altitude. Experiment 4 and 5 are completely the same, except for the addition of the trap tree in experiment 5. I would therefore suggest omitting experiment 4 from the results. Regarding experiment 1, this was only done on one location. Therefore, the variability between sites are not taken into account and no robust conclusions can be made.

I suggest reanalyzing the experiments. Did you check for multicollinearity between variables? Especially diameter and volume of the trap tree should be correlated. Including both variables is highly likely inflating the estimates and can even cause change in the direction of the estimate of any of the variables. Additionally, the authors used GLM for the analysis. I suggest that they use a generalized linear mixed model (GLMM). In which the location and year are used as random effect. In experiment 1 and 2, a GLM with binomial distribution was used. I do not understand why this was the case. As far as I read, the number of I. cembrae were taken into account in the GLM and would therefore need a Poisson distribution instead of a binonomial distribution. In experiment 1 it would be good also to include the side of the log as a variable in GLM model.

Discussion: I did not go into the discussion that much as this might change when new results would come up. Furthermore, many parts of the discussion are actually not discussing the results but have to do with the management implications in general.

Minor comments:

Abstract: does not contain all results and should therefore be longer

Introduction

Line 72-73: I do not think that this is the best way how to refer to your own experiences. Maybe refer that no research is known or something like that.

Materials and methods:

Lines 108-111: how did you decide for these sites? Please explain.

Results:

Tables with the statistics should include the estimates

Figure 7 does not show the correlation with altitude. (see caption)

Discussion:

388-397: I do not understand what this has to do with the results?

398-407: not discussing the results and promoting a certain product. Please delete the company name.

408-409: explain why it is not for mass trapping, but for monitoring

Author Response

Authors: Thank you for your thorough review and all helpful comments. We have improved the manuscript according to the recommendations or we comment on all points of the review below.

The manuscript is very confusingly written, and I do not know if this is due to the writing style or the setup of the experiments, but probably both. First regarding the writing style. The introduction is very long and does not seem to have a consistent structure. Furthermore, the authors want to show that Ips cembrae is potentially important pest, however, mentioning all the time that there are only local outbreaks which it not necessary kill the trees. Although I think it is important that a study for this species is important, I think the economic importance should be put in perspective. My suggestion is that the introduction, but also the discussion should be shortened and restructured.

Authors: thank you for this comment, we understand your opinion. We restructured and we hope - clarified this part of manuscript – and shortened them. We admit that this help improve manuscript a lot.

It would be very helpful that the experts show per experiment still the sub aim.

Authors: sub aim added at the beggining of every experiment (materials and methods chapter)

I also suggest that the discussion is divided into a part where the results and the management implications are discussed.

Authors: we restructured discussion of manuscript – and shortened it. We admit that this help improve manuscript a lot.

Then regarding the set up, I have the feeling that the authors did several independent experiments and now want to combine it in one manuscript, because not all results showed promising results to be a stand alone paper. First, many of the results of the experiments are overlapping. In most of the experiments volume and diameter are considered, although it was only explicitly tested for experiment 3.

Authors: We agree that the experiments overlap, so they are also summarized in one manuscript because in the overall comparison they give a comprehensive overview of recommendations for individual methods.

Furthermore, in experiment 3 also the altitude was included, but only 5 locations were sampled. To get a good result for altitude more locations should be considered, because this result could also because of other different factors in the separate locations. No good conclusions can be made regarding the altitude.

Authors: Thank you for this factual comment. Yes, if we wanted to analyze this topic better, more robust data would be needed. Therefore, we prefer to omit the altitude analysis from the manuscript and reduce the experiment.

Experiment 4 and 5 are completely the same, except for the addition of the trap tree in experiment 5. I would therefore suggest omitting experiment 4 from the results.

Authors: This is the only point of the review that we do not fully accept. We consider the result to be beneficial on its own, because in other species of bark beetles, pheromone traps generally catch more beetles (e.g. Lubojacký, Holuša 2011, 2013); in I. cembrae as the only bark beetle, our results prove that the catches are comparable.

Regarding experiment 1, this was only done on one location. Therefore, the variability between sites are not taken into account and no robust conclusions can be made.

Authors: We agree, the results of this experiment are subsequently and minorly discussed and no robust conclusions are drawn from them, but rather additional parameters to trap trees and logs.

I suggest reanalyzing the experiments. Did you check for multicollinearity between variables? Especially diameter and volume of the trap tree should be correlated. Including both variables is highly likely inflating the estimates and can even cause change in the direction of the estimate of any of the variables. Additionally, the authors used GLM for the analysis. I suggest that they use a generalized linear mixed model (GLMM). In which the location and year are used as random effect.

Authors: yes, this problem happened for some missunderstandment in data Exchange we removed in all the models the volume variable. Though most of the model variables did follow the previous behaviour. GLMM is a good idea even though we do not think it is not independent data (not independent) and there is no time to recalculate within these few days but that if it is convinced it must be so we would need more time for review.

In experiment 1 and 2, a GLM with binomial distribution was used. I do not understand why this was the case. As far as I read, the number of I. cembrae were taken into account in the GLM and would therefore need a Poisson distribution instead of a binonomial distribution.

Authors: Yes other reviewers also mentioned this, the reason for binomial was that the data was delivered not in counts of beetles, but in densities per dm2, we opted for binomial distribution for that purpose. 

In experiment 1 it would be good also to include the side of the log as a variable in GLM model.

Authors: We tried to include side in the model, but since no differences were found, it did not significantly change the model, so we kept the original model for simplicity.

Discussion: I did not go into the discussion that much as this might change when new results would come up. Furthermore, many parts of the discussion are actually not discussing the results but have to do with the management implications in general.

Authors: thank you for your comments, we restructured discussion of manuscript – and shortened it. We admit that this help improve manuscript a lot.

Minor comments:

Abstract: does not contain all results and should therefore be longer

Authors: we added more results in abstract

Introduction: Line 72-73: I do not think that this is the best way how to refer to your own experiences. Maybe refer that no research is known or something like that.

Authors: Deleted.

Materials and methods: Lines 108-111: how did you decide for these sites? Please explain.

Authors: added to study sites chapter: Based on willingness of forest owners and older data on sites with high I. cembrae abundance in the last 15 years[34], we selected 11 study sites and one transect (experiment 2), which were located across the entire territory of the Czech Republic and at elevations ranging from 320 to 680 m a.s.l. (Figure 1).

Results:

Tables with the statistics should include the estimates

Authors: we included estimates and std. errors in all tables.

Figure 7 does not show the correlation with altitude. (see caption)

Authors: Yes, if we wanted to analyze this topic better, more robust data would be needed. Therefore, according to other reviwers, we prefer to omit the altitude analysis from the manuscript and reduce the experiment.

Discussion:

388-397: I do not understand what this has to do with the results?

Authors: You are right, deleted

398-407: not discussing the results and promoting a certain product. Please delete the company name.  

Authors: Deleted

408-409: explain why it is not for mass trapping, but for monitoring

Authors: we added new explanation, but in general: The population density of bark beetles is not be substantially influenced by mass trapping with pheromone-baited traps, and forest hygiene was considered to be the most reliable long-term method against scolytid attacks (Dimitri et al.,1992). Even when considering higher trapping performance, enough beetles would remain untrapped to colonize susceptible hosts, quickly replenishing population density due to low intraspecific competition (Weber, 1987). The use of pheromone traps for mass trapping is no longer recommended for bark beetles. Only a small part of the local population is captured (estimated <20-30%, e.g. (Weslien, 1992; Weslien and Lindelöw, 1989) and it is recommended to apply one pheromone trap to each infested tree in the stand, which is very complicated in practice.

Round 2

Reviewer 1 Report

Thank you very much for the opportunity to re-review this manuscript. The authors are to be commended with the thoroughness with which they addressed reviewer comments. All of the comments from my review have been addressed adequately.

I have a few very minor suggestions on the English expression, listed below:

ln 27  since this paragraph has been added in, the scientific name for larch is now not given at first mention, it should be moved up here 

ln 29 replace "for ecological and sanitary" with "for ecological and phytosanitary reasons"

ln 58 the use of the word "bast" here is fine, as it is defined, but elsewhere throughout the manuscript you use "phloem", I would recommend replacing this word and always using phloem

ln 435 and 437 replace the word "needless" with "unnecessary"

Author Response

Response to Reviewer 1 Comments

Response: Thank you very much for your time and help to improve our manuscript.

Point 1: ln 27  since this paragraph has been added in, the scientific name for larch is now not given at first mention, it should be moved up here 

Response 1: We moved the scientific name from line 36 to line 27.

Point 2: ln 29 replace "for ecological and sanitary" with "for ecological and phytosanitary reasons"

Response 2: We changed this part, see line 29.

Point 3: ln 58 the use of the word "bast" here is fine, as it is defined, but elsewhere throughout the manuscript you use "phloem", I would recommend replacing this word and always using phloem

Response 3: As recommended, we changed the term "bast" to "phloem". Line 58.

Point 4: ln 435 and 437 replace the word "needless" with "unnecessary"

Response 4: In new version of manuscript we use only word “unnecessary”, lines 429-431.

Reviewer 3 Report

I am happy to see that the article has been improved considerably. There are only a few suggestions which I think would improve the manuscript even more.

I appreciate very much that the sub aims were included. I would only suggets that the subaims will be mentioned at the end of the introduction, after the aims.

I would still suggest to do the analysis with the GLMM with location and/or year as a random effect if necessary. The location or year should not be included in the model as a variable.

I am not convinced why the binomial distribution was used for a GLM. Maybe try to do GLM with gaussian distribution, but transform the dependent variable.

The authors do not agree with my comment regarding experiment 4 and 5. The main difference between this two experiment is that in experiment 4 the numbers were divided between males and females. Why was this not done for experiment 5? Furthermore, figure 8 and figure 10 have the same information regarding the comparison with pheromone traps and the tripods (only the trap trees is added in figure 10). This makes figure 8 redundent.

The authors took out the altitude from the model, but still left it in Figure 7. I would therefore suggest to prepare a new figure without the altitude.

Add in all the figures the statistical difference with letters or an asterisk.

Author Response

Response to Reviewer 3 Comments

Response: Thank you very much for your time. We tried to improve our manuscript according to your recommendations or at least explain our opinions.

Point 1: I appreciate very much that the sub aims were included. I would only suggets that the subaims will be mentioned at the end of the introduction, after the aims.

Response 1: The subaims are included in Introduction – after main aims. Lines: 80-90.

Point 2: I would still suggest to do the analysis with the GLMM with location and/or year as a random effect if necessary. The location or year should not be included in the model as a variable.

Response 2: You are right, we chose the GLMM with random effect of study sites (experiment 5; year of the research was the same) on your recommendation and corrected everything in the manuscript. We excluded study sites from the variable in the GLMM. The main results were not changed by the new analysis. Lines: 241-242; Table 9

Point 3: I am not convinced why the binomial distribution was used for a GLM. Maybe try to do GLM with gaussian distribution, but transform the dependent variable.

Response 3: We transformed the data using log transformation but there are several 0 values, which makes it difficult. We opted for adding constant of 1 to the data, though we are not really convinced of this step. Here are the results of the model for number of entry holes:

Deviance Residuals:

     Min        1Q    Median        3Q       Max 

-0.91806  -0.23121  -0.00355   0.24816   0.81524 

Coefficients:

              Estimate Std. Error t value Pr(>|t|)   

(Intercept)   0.168197   0.051189   3.286   0.0011 **

top   0.101731   0.039443   2.579   0.0103 * 

middle 0.098851   0.039443   2.506   0.0126 * 

diameter 0.028678   0.004337   6.613 1.18e-10 ***

---

Signif. codes:  0 ‘***’ 0.001 ‘**’ 0.01 ‘*’ 0.05 ‘.’ 0.1 ‘ ’ 1

(Dispersion parameter for gaussian family taken to be 0.1073488)

    Null deviance: 49.633  on 413  degrees of freedom

Residual deviance: 44.013  on 410  degrees of freedom

AIC: 256.95

Number of Fisher Scoring iterations: 2

In the model for number of maternal galleries the only difference which was found compared to our original model is that the upper position became significant:

Deviance Residuals:

     Min        1Q    Median        3Q       Max 

-1.43931  -0.28137   0.07588   0.38481   0.96049 

Coefficients:

              Estimate Std. Error t value Pr(>|t|)   

(Intercept)   0.506236   0.078698   6.433  3.5e-10 ***

top  0.219920   0.060640   3.627 0.000323 ***

middle 0.063981   0.060640   1.055 0.292002   

diameter 0.058317   0.006668   8.746  < 2e-16 ***

---

Signif. codes:  0 ‘***’ 0.001 ‘**’ 0.01 ‘*’ 0.05 ‘.’ 0.1 ‘ ’ 1

(Dispersion parameter for gaussian family taken to be 0.2537282)

Null deviance: 126.97  on 413  degrees of freedom

Residual deviance: 104.03  on 410  degrees of freedom

AIC: 613.06

Number of Fisher Scoring iterations: 2

We still believe that choosing a binomial distribution for our data was the right choice. The data is not an integer. The other two opponents agreed. And even if we understand your opinion, we would reserve the original analysis. With all due respect, if the use of binomial distribution in GLM is unacceptable to you, we can changed the results according to results of the attached analysis.

 Point 4: The authors do not agree with my comment regarding experiment 4 and 5. The main difference between this two experiment is that in experiment 4 the numbers were divided between males and females. Why was this not done for experiment 5? Furthermore, figure 8 and figure 10 have the same information regarding the comparison with pheromone traps and the tripods (only the trap trees is added in figure 10). This makes figure 8 redundent

Response 4: We understand your comment. Unfortunately, before the sex analysis was performed in experiment 5, it was partly destroyed during the transport of the material, so we do not have accurate data on sex, but only the total numbers from these traps. Therefore, it is important to keep the results of experiment 4. On the other hand, we recognize that Figure 8 is redundant and shows results identical to Figure 10. Therefore, we removed Figure 8 from the results.

Point 5: The authors took out the altitude from the model, but still left it in Figure 7. I would therefore suggest to prepare a new figure without the altitude.

Response 5: We added new version of Figure 7 without the altitude. See Figure 7.

Point 6: Add in all the figures the statistical difference with letters or an asterisk.

Response 6: We added statistical differences with asterisk in Figures 5, 8 and 9.
